# Blocking Bandits

**Soumya Basu**
UT Austin

**Rajat Sen**
Amazon

**Sujay Sanghavi**
UT Austin, Amazon

**Sanjay Shakkottai**
UT Austin

## Abstract

We consider a novel stochastic multi-armed bandit setting, where playing an arm makes it unavailable for a fixed number of time slots thereafter. This models situations where reusing an arm too often is undesirable (e.g. making the same product recommendation repeatedly) or infeasible (e.g. compute job scheduling on machines). We show that with prior knowledge of the rewards and delays of all the arms, the problem of optimizing cumulative reward does not admit any pseudo-polynomial time algorithm (in the number of arms) unless randomized exponential time hypothesis is false, by mapping to the PINWHEEL scheduling problem. Subsequently, we show that a simple greedy algorithm that plays the available arm with the highest reward is asymptotically $(1 - 1/e)$ optimal. When the rewards are unknown, we design a UCB based algorithm which is shown to have $c \log T + o(\log T)$ cumulative regret against the greedy algorithm, leveraging the free exploration of arms due to the unavailability. Finally, when all the delays are equal the problem reduces to Combinatorial Semi-bandits providing us with a lower bound of $c' \log T + \omega(\log T)$.

## 1 Introduction

We propose *Blocking Bandits* a novel stochastic multi armed bandits (MAB) problem where there are multiple arms with i.i.d. stochastic rewards and, additionally, each arm is *blocked* for a deterministic number of rounds. In online systems, such blocking constraints arise naturally when repeating an action within a time frame may be detrimental, or even be infeasible. In data processing systems, a resource (e.g. a compute node, a GPU) may become unavailable for a certain amount of time when a job is allocated to it. The detrimental effect is evident in recommendation systems, where it is highly unlikely to make an individual attracted to a certain product (e.g. book, movie or song) through incessant recommendations of it. A resting time between recommendations of identical products can be effective as it maintains diversity.

Surprisingly, this simple yet powerful extension of stochastic MAB problem remains unexplored despite the plethora of research surrounding the bandits literature [7, 1, 4, 8, 10] from its onset in [25]. Given the extensive research in this field, it is of no surprise that there are multiple existing ways to model this phenomenon. However, as we discuss such connections next, we observe that none of these approaches are direct, resulting in either large regret bounds or huge time complexity or both.

We briefly present the problem. There are $K$ arms, where mean reward $\mu_i$ is the reward and $D_i$ is the delay of arm $i$, for each $i = 1$ to $K$. When arm $i$ is played it is blocked for $(D_i - 1)$ time slots and becomes available on the $D_i$-th time slot after it's most recent play. The objective is to collect the maximum reward in a given time horizon $T$.

**Illustrative Example:** Consider three arms: arm 1 with delay 1 and mean reward $1/2$, arm 2 with delay 4 and mean reward 1, and arm 3 with delay 4 and mean reward 1. The reward maximization objective is met when the arms are played cyclically as $31213121\ldots$. There are two observations: First, due to blocking constraints we are forced to play *multiple arms* over time. Second, we note that the *order* in which arms are played is crucial. To illustrate, an alternate schedule $321 - 321 - \ldots$

('$-$' represents no arm is played) results in strictly less reward compared to the previous one as every fourth time slot no arm is available.

## 1.1 Main Contributions

We now present the main contributions of this paper.

1. **Formulation:** We formulate the blocking Bandits problem where each time an arm is played, it is blocked for a deterministic amount of time, and thus provides an abstraction for applications such as recommendations or job scheduling.

2. **Computational Hardness:** We prove that when the rewards and the delays are known, the problem of choosing a sequence of available arms to optimize the reward over a time horizon $T$ is computationally hard (see, Theorem 3.1). Specifically, we prove the offline optimization is as hard as *PINWHEEL Scheduling on dense instances* [18, 12, 20, 3], which does not permit any pseudo-polynomial time algorithm (in the number of arms) unless randomized exponential time hypothesis [5] is false.

3. **Approximation Algorithm:** On the positive side, we prove that the Oracle Greedy algorithm that knows the mean reward of the arms and simply plays the available arm with the highest mean reward is $(1 - 1/e - \mathcal{O}(1/T))$-optimal (see, Theorem 3.3). The approximation guarantee does not follow from standard techniques (e.g. sub-modular optimization bounds); instead it is proved by relating a novel lower bound of the Oracle Greedy algorithm to the LP relaxation based upper bound on MAXREWARD.

4. **Regret Upper Bound for UCB Greedy:** We propose the natural UCB Greedy algorithm which plays the available arm with the highest upper confidence bound. We provide regret upper bounds for the UCB Greedy as *compared to the Oracle Greedy* in Theorem 4.1.

Our proof technique is novel in two ways.

(i) In each time slot, the Oracle Greedy and the UCB Greedy algorithm have different sets of available arms (sample-path wise), as the set of available arms is correlated with the past decisions. We construct a coupling between the Oracle Greedy and the UCB Greedy algorithm, which enables us to capture the effect of learning error in UCB Greedy *locally in time* for each arm, despite the correlation with past decisions.

(ii) We prove that due to the blocking constraint, there is *free exploration* in the UCB Greedy algorithm. As the UCB Greedy algorithm plays the current *best* arm, it gets blocked, enforcing the play of the next *suboptimal* arm—a phenomenon we call free exploration. Free exploration ensures that upto a time horizon $t$, certain number of arms, namely $K^*$ (defined below), are played $ct$ amount of time each, for $c > 0$, w.h.p. Suppose $\mu_i$-s are non-decreasing with $i$. Let $K^* = \min\{i : \sum_{j=1}^{i} 1/D_i \geq 1\}$, and $\Delta(u, l) = \min\{\mu_i - \mu_{(i+1)} : l \leq i < u\}$. Then the regret is upper bounded by $\mathcal{O}(\frac{K(K-K^*)}{\Delta(K,K^*)} \log T)$. Ignoring free exploration the regret bound is $\mathcal{O}(\frac{K^2}{\Delta(K,1)} \log T)$, where $\Delta(K, 1) \leq \Delta(K, K^*)$.

5. **Regret Lower Bound:** We provide regret lower bounds for instances where the Oracle Greedy algorithm is optimal, and the regret is accumulated only due to learning errors. We consider the instances where all the delays are equal to $K^* < K$. We show under this setting the Oracle Greedy algorithm is optimal and the feedback structure of any online algorithm coincides with the combinatorial semi-bandit feedback [17, 13]. We show that for specific instances the regret admits a lower bound $\Omega(\frac{(K-K^*)}{\Delta(K,K^*)} \log T)$ in Theorem 4.3.

## 1.2 Connections to Existing Bandit Frameworks

We now briefly review related work in bandits, highlighting their shortcomings in solving the *stochastic blocking bandits* problem.

1. **Combinatorial Semi-bandits:** The blocking bandit problem is combinatorial in nature as the decisions of playing one arm changes the set of available arms in the future. Instead of viewing this problem on a per-time-slot basis, we can group a large block of time-slots together to determine a schedule of arm pulls and repeat this schedule, thus giving us an asymptotically optimal policy. We can now use ideas from stochastic Combinatorial semi bandits [13, 24] to learn the rewards by observing

all the rewards attained in each block. This approach, however, has two shortcomings. First we might need to consider extremely large blocks of time, specifically of size $\mathcal{O}(\exp(\text{lcm}(D_i : i \in [K] \log K))$ (lcm stands for the least common multiple), as an optimal policy may have periodic cycles of that length. This will require a large computational time as in the online algorithm the schedule will change depending on the reward estimates. Second, as the set of actions with large blocks is huge, the regret guarantees of such an approach may scale as $\mathcal{O}(\exp(\text{lcm}(D_i : i \in [K] \log K) \log T)$.

2. **Budgeted Combinatorial Bandits:** There are extensions to the above combinatorial semi bandit setting where additional *global budget constraints* are imposed, such as Knapsack constraints [26]— where an arm can only be played for a pre-specified number of times, and Budget constraints [28]— where each play of arm has an associated cost and the total expenditure has a budget. However, these settings cannot handle blocking that are *local* (per arm) in nature. Further, in [19] the authors consider adaptive adversaries, which can model our problem. But their approach will lead to a an approximation guarantee of $\mathcal{O}(1/\log(T))$ over $T$ timeslots.

An interesting recent work, Recharging Bandits [22] studies a system where the rewards of each arm is a concave and weakly increasing function of the time since the arm is played (i.e. a *recharging time*). However, the results therein do not apply as we focus on hard blocking constraints. Another work on bandits with delay-dependent payoffs [6] is not applicable as the results therein give no approximation guarantee for our setting.

3. **Sleeping Bandits:** Yet another bandit setting where the set of available actions change across time slots is Sleeping Bandits [23]. In this setting, the available action set is the same for all the competing policies including the optimal one in each time slot. However, in our scenario the set of available action in a particular time slot is dependent on the actions taken in the past time slots. Therefore, different policies may have different available action in each time slot. This precludes the application of ideas presented in Sleeping Bandits, and in sleeping combinatorial bandits [21], to our problem.

4. **Online Markov Decision Processes:** Finally, we can view this as a general Markov decision process on the state space $\mathcal{S} = [D_1] \times [D_2] \ldots [D_K]$, and the action space of arms $\mathcal{A} = [K]$, with mean reward $\mu_i$ for action $i$. The state space is again exponential in $K$, leading to huge computational bottleneck ($\mathcal{O}(\exp(K))$) and regret ($\mathcal{O}(\text{poly}(|S|) \log T)$) for standard approaches in online Markov decision processes [2, 27, 15].

## 2  Problem Definition

We consider a multi-armed bandit problem with blocking of arms. We have $K$ arms. For each $i \in [K]$, the $i$-th arm provides a reward $X_i(t)$ in time slot $t \geq 1$, where $X_i(t)$ are i.i.d. random variables with mean $\mu_i$ and support $[0, 1]$. Let us order the arms from highest to lowest reward w.l.o.g., s.t. $\mu_1 \geq \mu_2 \geq \cdots \geq \mu_K$. Let $\Delta_{ij} = \mu_i - \mu_j$ for all $1 \leq i < j \leq K$.

**Blocking:** For all $i \in [K]$, each arm $i$ is *deterministically blocked* for $(D_i - 1) \geq 0$ number of time slots once it is played. The actions of a player now decide the set of available arms due to blocking. In the $t$-th time slot, let us denote the set of available arms as $A_t$ and the arm pulled by the player as $I_t \in A_t$. For each $i \in [K]$, and $t \geq 1$, let the number of timeslots after and including $t$, the arm $i$ is blocked as $\tau_{i,t} = (D_i + \max_{t' \leq t}\{I_{t'} = i\} - t)$. The set of available arms at each time $t \geq$ is given as $A_t := A_t(i_1, \ldots, i_{t-1}) = \{\bar{i} : i \in [K], \tau_{i,t} \leq 0\}$. For a fixed time horizon, $T \geq 1$, the set of all valid actions is given as $\mathcal{I}_T = \{i_t \in A_t(i_1, \ldots, i_{t-1}) : t \in [T]\}$.

**Optimization:** Our objective is to attain the maximum expected cumulative reward. The expected cumulative reward of a policy $\mathbf{I}_T \in \mathcal{I}_T$ is given as $r(\mathbf{I}_T) = \mathbf{E}[\sum_{t=1}^T X_{i_t}(t)] = \sum_{i_t \in \mathbf{I}_T} \mu_{i_t}$. The offline optimization problem, with the knowledge of delays and mean rewards is stated as below.

$$\text{MAXREWARD: Solve OPT} = \max_{\mathbf{I}' \in \mathcal{I}_T} r(\mathbf{I}').$$

$\alpha$-**Regret:** We now define the $\alpha$-regret of a policy, which is identical to the $(\alpha, 1)$-regret defined in the combinatorial bandits literature [9]. For any $\alpha \in [0, 1]$, the $\alpha$-*regret of a policy* is the difference of expected cumulative reward of an $\alpha$-optimal policy and the expected cumulative reward of that policy, $R_T^\alpha = \alpha OPT - \mathbb{E}\left[\sum_{t=1}^T X_{i_t}(t)\right]$.

# 3 Scheduling with Known Rewards

## 3.1 Hardness of MAXREWARD

The offline algorithm is a periodic scheduling problem with the objective of reward maximization. In this section, we first prove (Corollary 3.2) that the offline problem does not admit any pseudo polynomial time algorithm in the number of arms, unless randomized exponential time hypothesis is false. We show hardness of the MAXREWARD problem by mapping it to the PINWHEEL SCHEDULING problem [18] as defined below.

**PINWHEEL SCHEDULING:** Given $K$ arms with delays $\{a_i : i \in [K]\}$, the PINWHEEL SCHEDULING problem is to decide if there exists a schedule (i.e. mapping $\Sigma : [T] \to [K]$ for any $T \geq 1$) such that for each $i \in [K]$ in $a_i$ consecutive time slots arm $i$ appears at least once.

We call such a schedule, if it exists, a valid schedule. A PINWHEEL SCHEDULING instance with a valid schedule is a YES instance, otherwise it is a NO instance. A PINWHEEL SCHEDULING instance is called *dense* if $\sum_{i=1}^{K} 1/a_i = 1$. Also, note that this problem is also known as *Single Machine Windows Scheduling Problem with Inexact Periods* [20].

**Theorem 3.1.** *MAXREWARD is at least as hard as PINWHEEL SCHEDULING on dense instances.*

In the proof, which is presented in the supplementary material, we show that given dense instances of PINWHEEL SCHEDULING there is an instance of MAXREWARD where the optimal value is strictly larger if the dense instance is an YES instance as compared to a NO instance. The following corollary provides hardness of MAXREWARD.

**Corollary 3.2.** *The problem MAXREWARD does not admit any pseudo-polynomial algorithm unless the randomized exponential time hypothesis is false.*

*Proof.* The proof follows from Theorem 3.1 and Theorem 24 in [20]. In [20], the authors shows that the PINWHEEL SCHEDULING with dense instances do not admit any pseudo-polynomial algorithm unless the randomized exponential time hypothesis [5] is False. ☐

## 3.2 $(1 - 1/e)$-Approximation of MAXREWARD

We study the *Oracle Greedy* algorithm where in each time slot the policy picks the best arm (i.e. the arm with highest mean reward $\mu_i$) in the set of available arms. We show in Theorem 3.3 that the greedy algorithm is $(1 - 1/e - \mathcal{O}(1/T))$ optimal[1] for the problem for any time-horizon $T$ and any number of arms $K$.

**Theorem 3.3.** *The greedy algorithm is asymptotically $(1 - 1/e)$ optimal for the MAXREWARD.*

**Proof Sketch:** The proof is presented in the supplementary material. It relies on three steps. Firstly, we show that using a Linear problem (LP) relaxation it is possible to obtain an upper bound to OPT in closed form as a function $f_{\text{upper}}(T, \mu_i, D_i, \forall i)$ of $\mu_i, D_i$ for all $i \in [K]$. In the next step, we show that the Greedy algorithm can be lower bounded as another function $f_{\text{lower}}(T, \mu_i, D_i, \forall i)$ of $\mu_i, D_i$ for all $i \in [K]$. The final step is to lower bound the ratio $\min_{\mu_i \in [0,1], D_i \geq 1, \forall i} \frac{f_{\text{lower}}(T, \mu_i, D_i, \forall i)}{f_{\text{upper}}(T, \mu_i, D_i, \forall i)}$. Our approach for the final step is to break this non-convex optimization into two steps, firstly optimization over $\mu_i$s which takes the form of a linear fractional program with a closed form lower bound as a function of $D_i, \forall i$. Secondly, we show that this value can be furthered lower bounded universally across all $D_i \geq 1, \forall i$, as $(1 - 1/e - \mathcal{O}(1/T))$.

## 3.3 Optimality Gap

We now show that greedy is suboptimal by constructing instances where greedy attains a cumulative reward $(3/4 - \delta)$ times the optimal reward, for any $\delta > 0$. Finally, the greedy algorithm that plays the available arm with maximum $\mu_i/D_i$ is shown to attain $1/K$ times the optimal reward in certain instances. We call this algorithm greedy-per-round.

**Proposition 3.4.** *For any $\epsilon > 0$, there exists an instance with $4$ arms where the greedy algorithm achieves $\frac{(3-\epsilon)}{4-2\epsilon}$ fraction of optimal reward.*

*Proof.* Consider the instance where arm 1 and 2 have reward 1 and delay 3, arm 3 has reward $1-\epsilon$ and delay 1, and arm 4 has reward 0 and delay 0. Also, each arm has only one copy. For any time horizon $T$ which is a multiple of 4, the greedy algorithm has the repeated schedule '1, 2, 3, 4, 1, 2, 3, 4, . . .'. Therefore, the reward for greedy is $(3-\epsilon)T/4$. Whereas, the optimal reward of $(4-2\epsilon)T/4$ is attained by the schedule '1, 3, 2, 3, 1, 3, 2, 3, . . .'. Therefore, the greedy achieves reward $\frac{(3-\epsilon)}{4-2\epsilon}$ times the optimal. $\square$

**Proposition 3.5.** *For any $\epsilon > 0$, there exists an instance with $K$ arms where the greedy-per-round algorithm achieves $\frac{(1+\epsilon)}{(K-2)}$ fraction of the optimal reward.*

*Proof.* Consider the instance where the arms 1 to $(K-1)$ each has reward 1, delay $(K-2)$. The $K$-th arm has reward $(1+\epsilon)/(K-2)$ for $\epsilon > 0$ and delay 0. The greedy-per-round will always play the $K$-th arm, as $(1+\epsilon)/(K-2) \geq 1/(K-2)$, attaining a reward of $(1+\epsilon)T/(K-1)$ in $T$ time-slots. Whereas, the optimal algorithm will play the arms $1, 2, \ldots, (K-1)$ in a round robin manner attaining a reward of $T$ in $T$ time-slots. Therefore, greedy-per-round can only attain $(K-2)/(1+\epsilon)$ fraction of the optimal reward. $\square$

# 4 Greedy Scheduling with Unknown Rewards

## 4.1 UCB Greedy Algorithm

In this section, we present the Upper Confidence Bound Greedy algorithm that operates without the knowledge of the mean rewards and the delays. The algorithm maintains the upper confidence bound for the mean reward of each arm, and in each time slot plays the available arm with the highest upper confidence bound, $\left(\hat{\mu}_i + \sqrt{\frac{8\log t}{n_i}}\right)$, where for arm $i$, $\hat{\mu}_i$ is the estimate of the mean reward and $n_i$ the total number of time arm $i$ has been played. [2]

---

**Algorithm 1** Upper Confidence Bound Greedy

1: **Initialize:** Mean estimate $\hat{\mu}_i = 0$ and Count $n_i = 0$, for all $i \in [K]$
2: **for all** $t = 1$ to $T$ **do**
3:     Play arm $i_t = \begin{cases} t, & \text{if} \quad t \leq K, \\ i_t = \arg\max_{i \in A_t}\left(\hat{\mu}_i + \sqrt{\frac{8\log t}{n_i}}\right), \text{o/w.} \end{cases}$
4:     **if** $i_t \neq \emptyset$ **then**
5:         $n_{i_t} \leftarrow n_{i_t} + 1$
6:         $\hat{\mu}_{i_t} \leftarrow \left(1 - \frac{1}{n_{i_t}}\right)\hat{\mu}_{i_t}(t) + \frac{1}{n_{i_t}}X_{i_t}(t).$

---

## 4.2 Analysis of UCB Greedy

We now provide an upper bound to the regret of the UCB Greedy algorithm as compared to the Oracle Greedy algorithm that uses the knowledge of the rewards. Let us recall that, the rewards are sorted (i.e. $\mu_i$ is non-increasing with $i$).

**Quantities used in Regret Bound.** $K_g$ is the worst arm with *mean reward strictly greater* than $0$ played by the Oracle Greedy algorithm. Let $H(m) = \sum_{n=1}^{\infty} 1/n^m, m > 1$ (Reimann zeta function).
We define $K_\epsilon^* = \min(K \cup \{k : \sum_{i=1}^{(k-1)} 1/D_k \geq 1 - \epsilon\})$ for any $\epsilon \geq 0$; and $K^* := K_0^*$.
For each $1 \leq k < k' \leq K$, let $\Delta(k, k') := \min\{\mu_i - \mu_j : i \leq k, j \geq k' + 1, i < j\}$.

Further for all $i = 1$ to $K_g$, and $j = (i+1)$ to $K_\epsilon^*$, we define $c_{ij} = \left(\frac{D_j}{\Delta_{ij}^2} + \frac{K}{\Delta_{j(j+1)}^2}\right)$. We note

$$\forall \epsilon \geq 0, \ (1-\epsilon)\min_i D_i \leq K_\epsilon^* \leq \min(K, (1-\epsilon)\max_i D_i + 1), \quad D_{min} \leq K_g \leq \min(K, D_{max}).$$

**Theorem 4.1.** *The $(1 - 1/e)$-Regret of UCB Greedy for a time horizon $T$ is upper bounded, for any $\epsilon > 0$, as*

$$\sum_{i=1}^{K_g} \left( 2H(4)\frac{\mu_i - \mu_K}{D_i^4} + H(3)K\frac{\mu_i - \mu_{K_\epsilon^*}}{D_i^3} + \sum_{j=(i+1)}^{K_\epsilon^*} \frac{\Delta_{ij}}{D_i}\frac{c_{ij}}{\epsilon}\log\left(\frac{c_{ij}}{\epsilon}\right) \right) + \sum_{i=1}^{K_g} \sum_{\substack{j=1+\\\max(i,K_\epsilon^*)}}^{K} \frac{32\log t}{\Delta_{ij}}.$$

**Simplified Regret Bound.** The regret admits the simplified upper bound for any $\epsilon > 0$,

$$R_T^{(1-1/e)} \leq \mathcal{O}\left(\frac{1}{\epsilon}\log\left(\frac{1}{\epsilon}\right)\right) + \frac{32K_g(K - K_\epsilon^*)}{\min_{i \in [K_\epsilon^*, \ldots, K_g]}\Delta_{i,i+1}}\log(T).$$

**Role of Free Exploration in Regret Bound.** Ignoring the free exploration in the system, we can upper bound the regret as $\sum_{i=1}^{K_g}\sum_{j=(i+1)}^{K}\frac{32\log(T)}{\Delta_{ij}} + 2H(4)\sum_{i=1}^{K_g}\frac{\mu_i - \mu_K}{D_i^4}$. Therefore, by capturing the free exploration, we are able to significantly improve the regret bound of the UCB Greedy algorithm when $\min\limits_{i<K_\epsilon^*}\Delta_{i(i+1)} << \min\limits_{i<K_\epsilon^*}\Delta_{i(K^*+1)}$.

*Proof Sketch of Theorem 4.1.* We present parts of the proof here, where the complete proof is deferred to the supplementary material. While computing the regret, we consider each arm $i = 1$ to $K_g$ separately. For each arm $i = 1$ to $K_g$, let $\mathcal{T}_i$ be the instances where greedy with full information, henceforth a.k.a. oracle Greedy (OG), plays arm $i$. Also, let $n_g(i) = |\mathcal{T}_i|$ be the number of time the greedy algorithm plays arm $i$. Let $X^g(t)$ be the *mean reward* obtained by OG in time slot $t$, which is a deterministic quantity. Recall, we denote the award obtained by UCB Greedy in time slot $t$ as $X_{i_t}(t)$, which is a random variable.

In the blocking bandit model, we end up with *free exploration* as each arm becomes unavailable for certain amount of time once it is played. This presents us with opportunity to learn more about the subsequent arms. However, when the delays, i.e. the $D_i$s, are arbitrary the OG algorithm itself follows a complicated repeating pattern, which is periodic but with period $\text{lcm}(D_i, i = 1$ to $K_g)$. We do not analyze the regret in a period directly, but consider the regret from each arm separately.

To understand our approach to regret bound, let us fix an arm $i \leq K_g$. We consider the time slots divided into blocks of length $D_i$, where each block begins at an instance where OG plays arm $i$. In each block, the arm $i$ becomes available at least once for any algorithm, including UCB Greedy (UCBG); but not necessarily at the beginning as in case of OG. In each such block, if we play arm less of equal to $i$ when it becomes first available we don't accumulate any regret when the reward from arm $i$ is considered in isolation. Instead, if we play arm $j \geq (i + 1)$ when arm $i$ becomes first available we may upper bound the regret as $\Delta_{ij}$ in that block. Let us denote by $P_{ij}(t)$ the probability that arm $j \geq (i + 1)$ is played in the block starting at time $t \in \mathcal{T}_i$ where arm $i$ becomes available first.

Using the previous logic, separately for each arm and using linearity of expectation we arrive at the following regret bound.

$$\sum_{i=1}^{T} X^g(t) - \mathbb{E}\left[\sum_{i=1}^{T} X_{i_t}(t)\right] \leq \sum_{i=1}^{K_g}\sum_{t \in \mathcal{T}_i}\sum_{j=(i+1)}^{K} P_{ij}(t)\Delta_{ij}. \tag{1}$$

While bounding the regret in equation 7, in order to account for the combinatorial constraints due to the unavailability of arms, we phrase it as the following optimization problem (9).

$$\max \sum_{i=1}^{K_g}\sum_{t \in \mathcal{T}_i}\sum_{j=(i+1)}^{K} P_{ij}(t)\Delta_{ij} \tag{2}$$

$$s.t.\ P_{ij}(t) \leq \frac{2}{t^4} + \mathbb{P}\left(n_j(t) \leq \frac{32\log t}{\Delta_{ij}^2}; a_t = j\right)\left(1 - \frac{2}{t^4}\right), \forall i, j \in [K], \qquad \forall t \in \mathcal{T}_i, \tag{3}$$

$$n_j(t) \geq c_j t - c_j'\log t,\ \text{w.p.} \geq (1 - K/t^3), c_j, c_j' > 0 \qquad \forall j \leq K_\epsilon^*, \tag{4}$$

The first constraint is standard, whereas the second constraint represent the *free exploration* in the system. If any arm $i$ is played $n_i(t)$ times upto time $t$ then it is available for $(t - n_i(t)D_i)$ time slots.

Among these time slots where arm $i$ is available, UCBG can play

1) arms $1 \le j \le (i-1)$, at most $\sum_{j=1}^{(i-1)} \left( \frac{t}{D_j} + 1 \right)$ times in total, w.p. 1, due to the blocking constraints; and

2) the arms $(i + 1) \le j \le K$, can be played at most $\sum_{j=(i+1)}^{K} \frac{32 \log t}{\Delta_{ij}^2}$ many times in total, w.p. at least $(1 - K/t^3)$, due to the UCB property and union bound over all arms and time slots upto $t$.

Therefore, for all $i \le K$ we have, w.p. at least $(1 - K/t^3)$,

$$n_i(t) \ge \frac{t}{D_i} \left( 1 - \sum_{j=1}^{(i-1)} \frac{1}{D_j} \right) - \frac{1}{D_i} \left( \sum_{j=(i+1)}^{K} \frac{32 \log t}{\Delta_{ij}^2} + (i-1) \right). \tag{5}$$

More importantly, w.h.p. for all $i \le K_\epsilon^*$ we see $n_i(t)$ grows linearly with time $t$. This provides us with the required upper bound after using the lower bounds for $n_j(t)$ for $j = 1$ to $K_\epsilon^*$, appropriately. $\quad\square$

### 4.3 Easy Instances and Regret Lower Bound

In this section, we show that there are class of instances where Oracle greedy is optimal and provide regret lower bounds for such a setting.

**Definition 4.2.** *An instance of the blocking bandit is an* easy instance *if the Oracle Greedy is an offline optimal algorithm for that instance.*

*Examples:* 1) A class of examples of such easy instances is blocking bandits where all the arms have equal delay $D < K$.
2) When the sequences $\text{seq}_i := \{i + kD_i : k \in \mathbb{N}\}$ for $i = 1$ to $K_g$ do not collide in any location $(\text{seq}_i \cap \text{seq}_j = \emptyset, \forall i \ne j)$ and cover the integers $\forall T \ge 1, [T] \subseteq \cup_{i=1}^{K_g}\text{seq}_i$ ( a.k.a. exact covering systems [16]) then Oracle Greedy is asymptotically optimal.

**Lower Bound:** We now provide a lower bound on the regret for easy instances . An algorithm is *consistent* iff for any *instance* of stochastic blocking bandit, the regret upto time $T$, $R_T^1 = o(T^\delta)$ for all $\delta > 0$. We prove the regret lower bound over the class of *consistent* algorithms for *easy instances* of stochastic blocking bandits.

We consider an instance with equal delay $D < K$, which is an *easy instance*. In this instance, the rewards for each arm $i = 1$ to $K^*$ has Bernoulli distribution with mean $1/2$; whereas arms $i = (K^* + 1)$ to $K$ has reward $(1/2 - \Delta)$. We call this instance $K^*$-Set and prove the following.

**Theorem 4.3.** *For any $K$ and $K^* < K$ and $\Delta \in (0, 1/2)$ the regret of any* consistent *algorithm on the $K^*$-Set instance is lower bounded as* $\lim_{T\to\infty} \frac{R_T^1}{\log T} \ge \frac{(K - K^*)}{\Delta}$.

The proof of the above theorem makes use of the following lemma which shows that the blocking bandit instance is equivalent to that of a combinatorial semi-bandit [11], problem on $m$-sets, for which regret lower bounds were established in [1].

**Lemma 4.4.** *For any Blocking Bandit instance where $D_i = D \le K$ for all arms $i \in [K]$, time horizon $T$, and any online algorithm $\mathcal{A}_O$, there exists an online algorithm $\mathcal{A}_B$ which chooses arms for blocks of $D$ time slots and obtain the same distribution of the cumulative reward as $\mathcal{A}_O$.*

The proof of the lemma is deferred to the supplementary material.

## 5 Experimental Evaluation

**Synthetic Experiments:** We first validate our results on synthetic experiments, where we use $K = 20$ arms. The gaps in mean rewards of the arms are fixed with $\Delta_{i(i+1)}$, chosen uniformly at random (u.a.r.) from $[0.01, 0.05]$ for all $i = 1$ to $19$. We also fix $\mu_K = 0$. The rewards are distributed as Bernoulli random variables with mean $\mu_i$. The delays are fixed either 1) by sampling all delays u.a.r. from $[1, 10]$ (small delay instances), or 2) u.a.r. from $[11, 20]$ (large delay instances), or 3) by fixing all the delay to a single value.

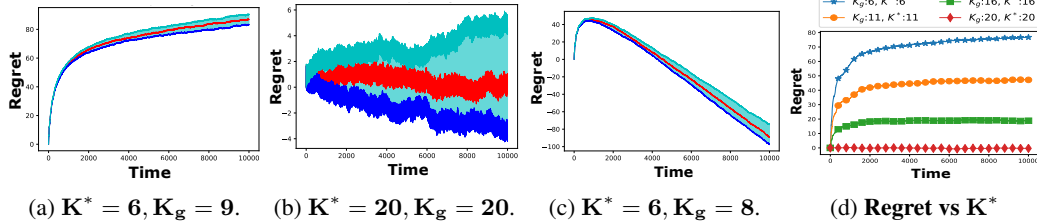

(a) $\mathbf{K^* = 6, K_g = 9}$.    (b) $\mathbf{K^* = 20, K_g = 20}$.    (c) $\mathbf{K^* = 6, K_g = 8}$.    (d) **Regret vs $\mathbf{K^*}$**

Figure 1: Cumulative regrets scale as logarithmic, constant, and negative linear regret with randomly initialized delays, in Fig.1a, Fig.1b, and Fig.1c, resp. Fig.1d: Regret vs $K^*$ with identical delays.

Once the rewards and the delays are fixed, we run both the oracle greedy and the UCB Greedy algorithm 250 times to obtain the expected regret (i.e. Reward of Oracle Greedy - Reward of UCB Greedy) trajectory each with $10k$ timeslots. For each setting, we repeat this process 50 times for each experiment to obtain 50 such trajectories. We then plot the median, $75\%$ and $25\%$ points in each timeslot accorss all these 50 trajectories in Figure 1.

*Scaling with Time:* We observe three different behaviors. In most of the cases, we observe the regret scales logarithmically with $T$ (see, Fig. 1a). In the second situation, when $K^* = K_g$ the typical behavior is depicted in Fig. 1b where we observe constant regret (for $K^* = K$ the logarithmic part vanishes in our regret bounds). Finally, there are instances, as shown in Fig.1c, when the regret is negative and scales linearly with time. Note as the Oracle greedy is suboptimal UCB Greedy can potentially outperform it and have negative regret. As an example consider the *illustrative example* in Section 1. In this example, if due to learning error the UCB greedy plays the sequence '121' then the UCB Greedy gets latched to the sequence '12131213...'—which is optimal. Such events can happen with constant probability, resulting in a reward linearly larger than the Oracle Greedy which plays '$321 - 321 - \dots$'. This example explains the instances with linear negative regret.

*Scaling with $K^*$:* In Fig.1d, (where only the median is plotted) we consider the instances with identical delay equal to $K^* = 7, 11, 16, 20$. We observe that the regret decreases with increasing $K^*$, which is similar to the proved lower bound.

**Jokes Recommendation Experiment:** We perform jokes recommendation experiment using the Jesters joke dataset [14]. In particular, we consider 70 jokes from the dataset, each joke with at least $15k$ valid ratings in range $[-10, 10]$. We rescale the ratings to $[0, 1]$ using $x \rightarrow (x + 10)/20$. In our experiments, when a specific joke is recommended a rating out of the more than $15k$ ratings is selected uniformly at random with repetition and this rating acts as the instantaneous reward. The task is to recommend jokes to maximize the rating over a time horizon, with blocking constraints for each joke. The delays are chosen randomly similar to the synthetic experiments. For each experiment, we plot the expected regret trajectory for $15k$ time slots, taking expectation over 500 simulated sample paths. We observe the expected scaling behavior, where the regret scales logarithmically in time and for larger $K^*$ we observe smaller regret.

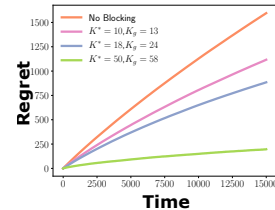

Figure 2: Regret vs $K^*$ in jokes recommendation with blocking.

## 6 Conclusion

We propose *blocking bandits*, a novel stochastic multi-armed bandit problem, where each arm is blocked for a specific number of time slots once it is played. We provide hardness results and approximation guarantees for the offline version of the problem, showing an online greedy algorithm provides an $(1 - 1/e)$ approximation. We propose UCB Greedy and analyze the regret upper bound through novel techniques, such as *free exploration*. For instances on which oracle greedy is optimal we provide lower bounds on regret. Improving regret bounds using the knowledge of the delays of the arms is an interesting future direction which we intend to explore. In another direction, providing better lower bounds through novel constructions (e.g. exact covering systems) can be investigated.

**Acknowledgements:** This research was partially supported by NSF Grant 1826320, ARO grant W911NF-17-1-0359, the Wireless Networking and Communications Group Industrial Affiliates Program, and the the US DoT supported D-STOP Tier 1 University Transportation Center.

## Footnotes

[1]An algorithm is $\alpha$ optimal for the offline problem if the expected cumulative reward is $\alpha$ times the optimal expected cumulative reward

[2]We believe with some increased complexity in the proof, the constant 8 in UCB can be improved to 2.

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
