[Supplementary Material]

# A    Proof of Hardness of MAXREWARD, Theorem 3.1

Given a dense PINWHEEL SCHEDULING instance $\{a_i : i \in [K]\}$ we construct a MAXREWARD instance. For each $i \in [K]$, we have an arm with delay $a_i$ and reward 1. Additionally, we have an arm $(K+1)$ which has delay 0 and reward 0.

Case 1: The PINWHEEL SCHEDULING instance is a YES instance, i.e. there exists a valid schedule with arms $i \in [K]$. Furthermore, as the instance is a dense instance we have exact $a_i$ period for each arm $i \in [K]$. It also means for all $T \geq 1$ there is no empty slot in the schedule. This implies that pulling the arms according to the above schedule we obtain a valid solution for MAXREWARD with cumulative reward $T$ in $T$ time slots, for any $T \geq 1$.

Case 2: The PINWHEEL SCHEDULING instance is a NO instance, i.e. there does not exist a valid schedule with arms $i \in [K]$. This implies for any schedule, there exists a block of $a_i$ time slots such that arm $i$ is not scheduled in that block, for some $i \in [K]$. However, as the instance is dense it implies that there exists a gap in any schedule. This in turn implies that in the MAXREWARD problem any valid solution has to (in the afore mentioned gap) play the $(K+1)$-th arm at least once. Coupled with the fact that any schedule for is periodic with period $\prod_{i=1}^{K} a_i$, this implies that for $T \geq 1$ we can obtain at most $\left(T - \lfloor T/\prod_{i=1}^{K} a_i \rfloor\right)$.

For $T$ large enough there is a non-zero gap in the reward obtained in Case 1 and Case 2. Therefore, by solving the above MAXREWARD instance, we can decide whether the PINWHEEL SCHEDULING instance is a YES instance or a NO instance.    $\square$

**Remark** We note that the $T$ in the above reduction is expressible in polynomial many bits (w.r.t. number of arms) as taking $T = 100\prod_{i=1}^{K} a_i$ suffices, where $a_i$ are themselves expressible in polynomial many bits. Thus the "hard" decision problem for MAXREWARD is $OPT = T$?.

# B    Proof of (1-1/e)-Approximation of MAXREWARD, Theorem 3.3

For the purpose of the proof assume $T$ is an arbitrary fixed integer.

**ILP formulation:** The problem of max reward scheduling can be formulated as the following integer program, with the interpretation that $x_{k,t} = 1$ if and only if the arm $k$ is chosen at time $t$, for all $k \in [K]$ and $t \in [T]$.

$$\max_{x_{k,t}} \sum_{t \in [T]} \sum_{k \in [K]} x_{k,t} \mu_k$$

$$s.t.1)\ x_{k,t} \in \{0,1\} \forall k \in [K], t \in [T]; 2) \sum_{k \in [K]} x_{k,t} \leq 1, \forall t \in [T];$$

$$3) \sum_{t \in [D_k]} x_{k,t+t_0} \leq 1, \forall t_0 \in [T+1-D_k], \forall k \in [K]$$

**LP Upper Bound:** We can obtain an upper bound for the above integer program using the following linear program (LP), with the interpretation that $n_k$ is the number (possibly fractional) of time slots the arm $k$ is played. This is obtained by relaxing the conditions in 1 to $x_{k,t} \in [0,1]$.

$$\max_{n_k} \sum_{k \in [K]} n_k \mu_k; \quad s.t.1)\ n_k \in [0, \lceil T/D_k \rceil], \forall k \in [K]; 2) \sum_{k \in [K]} n_k = T$$

The above LP admits the solution, $\forall k \in [K], n_k^* = \min\left\{\lceil T/D_k \rceil, \left(T - \sum_{i=1}^{k-1} \lceil T/D_i \rceil\right)^+\right\}$. Let $K^* \leq K$ be the highest arm with non-zero $n^*k$.

**Lower Bound on Greedy Algorithm:** We now lower bound the reward collected by the greedy algorithm. Let $n_k^g$ be the number of times arm $k$ is pulled under the greedy algorithm. Let us denote the time slots occupied by arm 1 to $k$ under greedy schedule as $sch_1^k$. The time slots, where the periodic placement of arm $i$ collides with already placed arms 1 to $(i-1)$ is denoted

as $col_i = \{t : i \leq t \leq T, D_i | (t - i), t \in sch_1^{(i-1)}\}$. Then the number of time arm $i$ is played is $\lceil (T - (|col_i| + i - 1))/D_i \rceil$. This holds because for arm $i$ we can remove the time-slots with collisions along with the initial $(i - 1)$ timeslots, and perform periodic placement perfectly with remaining $(T - (|col_i| + i - 1))$ time slots. We note that $(|col_i| + i - 1) \leq \sum_{j=1}^{i-1} n_j^g$.

We now define for each $k \in [K]$, $n_k' = T_k/D_k$, and $T_k = \left(T - \sum_{j=1}^{k-1} n_j'\right)^+$. The interpretation is that iteratively we remove the timeslots where the previous arms 1 to $i$ are placed and then place arm $i$ periodically with period $D_i$. Our claim is that for all $k$, $\sum_{i=1}^{k} n_i^g \geq \sum_{i=1}^{k} n_i'$. This claim immediately implies that $\sum_{k \in [K]} n_k^g \mu_k \geq \sum_{k \in [K]} n_k' \mu_k$ as the rewards are sorted non-decreasingly with $k$. We prove the claim using induction on $k$. We know that $n_1^g \lceil T/D_1 \rceil \geq n_1'$. By induction hypothesis, we suppose $\sum_{i=1}^{k} n_i^g \geq \sum_{i=1}^{k} n_i'$ for all $k \leq (k' - 1)$. We have

$$n_{k'}^g = \lceil (T - (|col_{k'}| + k' - 1))/D_{k'} \rceil \geq \frac{1}{D_{k'}} \left(T - \sum_{i=1}^{k'-1} n_i^g\right)$$

$$\geq \frac{1}{D_{k'}} \left(T - \sum_{i=1}^{k'-1} n_i' - \sum_{i=1}^{k'-1} (n_i^g - n_i')\right) = n_{k'}' - \frac{1}{D_{k'}} \sum_{i=1}^{k'-1} (n_i^g - n_i').$$

Therefore, $\sum_{i=1}^{k'} (n_i^g - n_i') \geq (1 - 1/D_{k'}) \sum_{i=1}^{k'-1} (n_i^g - n_i')$, which means $\sum_{i=1}^{k'} n_i^g \geq \sum_{i=1}^{k'} n_i'$. The induction hypothesis is proved.

Finally, we note that for each $k \in [K]$, $n_k' = \frac{T}{D_k} \prod_{i=1}^{k-1} \left(1 - \frac{1}{D_i}\right)$ which can be shown easily using induction over $k$.

**Greedy Lower Bound vs LP Upper Bound:** Finally, we note that the approximation guarantee of the greedy algorithm is given as follows, where $\frac{1}{\tilde{D}_{K^*}} = \left(1 - \sum_{i=1}^{K^*-1} \frac{1}{D_i}\right)$.

$$\frac{\sum_{k \in [K]} n_k^g \mu_k}{\sum_{k \in [K]} n_k^* \mu_k} \geq \frac{\sum_{k \in [K]} \frac{\mu_k}{D_k} \prod_{i=1}^{k-1} \left(1 - \frac{1}{D_i}\right)}{\sum_{k=1}^{K^*-1} \frac{\mu_k}{D_k} + \frac{\mu_{K^*}}{\tilde{D}_{K^*}}} \left(1 + \frac{D_1 K}{T \mu_1}\right)^{-1}.$$

We want to lower bound the following uniformly over all feasible $D_k$ and $\mu_k$ to prove our approximation guarantee.

$$\min \left(\sum_{k \in [K]} \frac{\mu_k}{D_k} \prod_{i=1}^{k-1} \left(1 - \frac{1}{D_i}\right)\right) \left(\sum_{k=1}^{K^*-1} \frac{\mu_k}{D_k} + \frac{\mu_{K^*}}{\tilde{D}_{K^*}}\right)^{-1}, \tag{6}$$

$$\text{s.t. } \forall i, j \in [K], i < j; \ (1) \ \mu_j \leq \mu_i, \ (2) \ \mu_i \in [0, 1], \ (3) \ D_i \geq 1.$$

We break the minimization into two steps, where we first minimize over $\mu_i$ as a function of $D_i$. Next we minimize over $D_i$.

**Part I:** In the first minimization, any optimal solution will have $\mu_k = 0$ for all $k \geq (K^* + 1)$. Otherwise, we can strictly decrease the objective. Next, to eliminate the inequalities among $\mu_i$s we make the substitution, $\mu_i = \mu_{(i-1)} - z_i \mu_1 = (1 - \sum_{j=2}^{i} z_j)\mu_1$, for all $i = 2$ to $K^*$. Also, for notational convenience denote $P_i = \prod_{j=1}^{i}(1 - 1/D_i)$ and $S_i = \sum_{j=1}^{i} 1/D_j$, for $i = 1$ to $K^*$. In the denominator we have,

$$\mu_1/D_1 + \sum_{k=2}^{K^*-1} \frac{\mu_1(1 - \sum_{j=2}^{k} z_j)}{D_k} + \frac{\mu_1(1 - \sum_{j=1}^{K^*} z_j)}{\tilde{D}_{K^*}}$$

$$= \mu_1 - \mu_1 \sum_{j=2}^{K^*} z_j \left(\sum_{k=j}^{K^*-1} 1/D_K + 1/\tilde{D}_{K^*}\right) = \mu_1 - \mu_1 \sum_{j=2}^{K^*} z_j (1 - S_{j-1}).$$

Similarly, in the numerator we have (after setting $\mu_k = 0$ for all $k \geq (K^* + 1)$),

$$\sum_{k=1}^{K^*} \frac{\mu_1(1 - \sum_{j=2}^{k} z_j)}{D_k} \prod_{i=1}^{k-1}\left(1 - \frac{1}{D_i}\right) = \mu_1 \sum_{k=1}^{K^*} \frac{1}{D_k} \prod_{i=1}^{k-1}\left(1 - \frac{1}{D_i}\right) - \mu_1 \sum_{j=2}^{K^*} z_j \sum_{k=j}^{K^*} \frac{1}{D_k} \prod_{i=1}^{k-1}\left(1 - \frac{1}{D_i}\right)$$

$$= \mu_1(1 - P_{K^*}) - \mu_1 \sum_{j=2}^{K^*} z_j(1 - P_{K^*} - 1 + P_{j-1}) = \mu_1(1 - P_{K^*}) - \mu_1 \sum_{j=2}^{K} z_j(P_{j-1} - P_{K^*}).$$

With the substitution, in the first stage we require to solve the following linear fractional optimization,

$$\min \frac{(1 - P_{K^*}) - \sum_{i=2}^{K^*} z_i \left(P_{(i-1)} - P_K\right)}{1 - \sum_{i=2}^{K^*} z_i(1 - S_{(i-1)})}, \text{ s.t. (1) } \forall i \geq 2, z_i \geq 0, \text{ (2) } \sum_{i=1}^{K^*} z_i \leq 1.$$

Through standard transformation to linear program we obtain an equivalent formulation of

$$\min \ (1 - P_{K^*}) + \sum_{i=2}^{K^*} y_i(1 - P_{(i-1)} - (1 - P_K)S_{(i-1)}), \quad \text{s.t. (1) } \forall i \geq 2, y_i \geq 0, \text{ (2) } \sum_{i=1}^{K^*} y_i \leq 1.$$

The above optimization admits a closed form solution with the value

$$(1 - P_{K^*}) + \min\left(0, \ \min_{i=1 \text{ to } K^*}(1 - P_{(i-1)} - (1 - P_{K^*})S_{(i-1)})\right).$$

We now prove that $\min_{i=1 \text{ to } K^*}(1 - P_{(i-1)} - (1 - P_{K^*})S_{(i-1)}) \geq 0$. We fix an $2 \leq i \leq (K^* - 1)$.

$$P_{K^*} = P_i \prod_{j=i+1}^{K^*} (1 - 1/D_i) \overset{i}{\geq} P_i(1 - \sum_{j=i+1}^{K^*} 1/D_i) = P_i S_i$$

$$\implies (1 - P_i - (1 - P_{K^*})S_i)$$

$$\geq (1 - P_i - (1 - P_i S_i)S_i) = (1 - S_i)(1 - P_i(1 + S_i))$$

$$\overset{(ii)}{\geq} (1 - S_i)\left(1 - P_i \prod_{j=1}^{i}(1 + 1/D_i)\right) = (1 - S_i)\left(1 - \prod_{j=1}^{i}(1 - 1/D_i^2)\right) \geq 0.$$

In the above we use Weierstrass' Inequality[3] in (i) and (ii). This concludes that first part of the optimization function results in $(1 - P_{K^*})$.

**Part II:** In the second part, we need to solve the following optimization problem.

$$\min\left(1 - \prod_{i=1}^{K^*}(1 - 1/D_i)\right), \text{ s.t. } \sum_{i=1}^{K^*} 1/D_i \geq 1, \forall i, D_i \geq 1.$$

From the first order KKT conditions of the above optimization we have,
(1) For all $i$, $\lambda - \prod_{j=1, j\neq i}^{K^*}(1 - 1/D_j) = 0$, and (2) $\lambda \geq 0 \implies \sum_{i=1}^{K^*} 1/D_i = 1$.
As for all $i$, $\prod_{j=1, j\neq i}^{K^*}(1 - 1/D_j) \geq 0$ we must have $D_i = K^*$ in the optimum solution. Therefore, the previous optimization problem admits the optimal value $\left(1 - (1 - 1/K^*)^{K^*}\right)$. This further implies that universally we have the lower bound $(1 - 1/e - KD_1\mu_1^{-1}/T)$ for the optimization problem (6). We conclude that the Greedy algorithm is an asymptotically $(1 - 1/e)$ approximation of the MAXREWARD problem.

## C    Proof of Regret Upper Bound, Theorem 4.1

In this section we first prove a theorem which is a slightly different from Theorem 4.1, and then show how to obtain Theorem 4.1.

**Theorem C.1.** *The regret of UCB Greedy algorithm to Greedy algorithm in time horizon $T$ is bounded from above by*

$$\sum_{i=1}^{K_g}\left(2H(4)\frac{\mu_i-\mu_K}{D_i^4}+H(3)K\frac{\mu_i-\mu_{K^*}}{D_i^3}+\sum_{j=(i+1)}^{K^*}\frac{\Delta_{ij}\tau_{ij}}{D_i}\right)+\sum_{i=1}^{K_g}\sum_{j=(K^*+1)}^{K}\frac{32\log t}{\Delta_{ij}},$$

*where for all $(i,j)$, $j\leq K^*$, and $i<j$, $\tau_{ij}\leq\tau_{(ij,0)}(1+\log\tau_{(ij,0)})+\tau_{(ij,1)}$,*

$$\tau_{(ij,0)}=32\left(1-\sum_{l=1}^{(j-1)}\frac{1}{D_l}\right)^{-1}\left(\frac{D_j}{\Delta_{ij}^2}+\sum_{l=(j+1)}^{K}\frac{1}{\Delta_{jl}^2}\right),\quad\tau_{(ij,1)}=\left(1-\sum_{l=1}^{(j-1)}\frac{1}{D_l}\right)^{-1}(j-1).$$

*Proof.* While computing the regret, we consider each arm $i=1$ to $K_g$ separately. For each arm $i=1$ to $K_g$, let $\mathcal{T}_i$ be the instances where greedy with full information, henceforth a.k.a. oracle Greedy (OG), plays arm $i$. Also, let $n_g(i)=|\mathcal{T}_i|$ be the number of time the greedy algorithm plays arm $i$. Let $X^g(t)$ be the *mean reward* obtained by OG in time slot $t$, which is a deterministic quantity. Recall, we denote the award obtained by UCBG in time slot $t$ as $X_{i_t}(t)$, which is a random variable.

In the blocking bandit model, we end up with *forced exploration* as each arm becomes unavailable for certain amount of time once it is played. This presents us with opportunity to learn more about the subsequent arms. However, when the delays, i.e. the $D_i$s, are arbitrary the OG algorithm itself follows a complicated repeating pattern, which is periodic but with period $\mathrm{lcm}(D_i, i=1\,to\,K_g)$. We do not analyze the regret in a period directly, but consider the regret from each arm separately.

To understand our approach to regret bound, let us fix an arm $i\leq K_g$. We consider the time slots divided into blocks of length $D_i$, where each block begins at an instance where OG plays arm $i$. In each block, the arm $i$ becomes available at least once for any algorithm, including UCB Greedy (UCBG); but not necessarily at the beginning as OG. In each such block, if we play arm $i$ when it becomes first available we don't accumulate any regret when the reward from arm $i$ is considered in isolation. Instead, if we play arm $j\geq(i+1)$ when arm $i$ becomes first available we may upper bound the regret as $\Delta_{ij}$ in that block. Let us denote by $P_{ij}(t)$ the probability that arm $j\geq(i+1)$ is played in the block starting at time $t\in\mathcal{T}_i$ where arm $i$ becomes available first.

Using the previous logic, separately for each arm and using linearity of expectation we arrive at the following regret bound.

$$\sum_{i=1}^{T}X^g(t)-\mathbb{E}\left[\sum_{i=1}^{T}X_{i_t}(t)\right]\leq\sum_{i=1}^{K_g}\sum_{t\in\mathcal{T}_i}\sum_{j=(i+1)}^{K}P_{ij}(t)\Delta_{ij}. \tag{7}$$

For our analysis, we require the following standard guarantee about the confidence intervals under UCB algorithm as given in[23], which follows from the application of Chernoff-Hoefding bound.

**Lemma C.2** ([23]). *For the random variables $n_i(t)$ and $\hat{\mu}_i$ in Algorithm 1, the following holds for all arms $1\leq i\leq K$ and for all time slots $1\leq t\leq T$,*

$$\mathbb{P}\left[\mu_i\notin\left[\hat{\mu}_i-\sqrt{\frac{8\ln t}{n_i(t)}},\hat{\mu}_i+\sqrt{\frac{8\ln t}{n_i(t)}}\right]\right]\leq\frac{1}{t^4}. \tag{8}$$

While bounding the regret in equation 7, in order to account for the combinatorial constraints due to the unavailability of arms, we phrase it as the following optimization problem (9).

$$\max\sum_{i=1}^{K_g}\sum_{t\in\mathcal{T}_i}\sum_{j=(i+1)}^{K}P_{ij}(t)\Delta_{ij} \tag{9}$$

$$s.t.\ P_{ij}(t)\leq\frac{2}{t^4}+\mathbb{P}\left(n_j(t)\leq\frac{32\log t}{\Delta_{ij}^2};a_t=j\right)\left(1-\frac{2}{t^4}\right),\forall i,j\in[K],\qquad\forall t\in\mathcal{T}_i,, \tag{10}$$

$$n_j(t)\geq c_j t-c_j'\log t,\ \text{w.p.}\ \geq(1-K/t^3),c_j,c_j'>0\qquad\qquad\forall j\leq K^*, \tag{11}$$

**Correctness of Optimization** (9).

- Eq. (10) holds due to Lemma C.2. We prove it as follows.

$$P_{ij}(t) \leq \mathbb{P}\left[\hat{\mu}_i + \sqrt{\frac{8\log t}{n_i(t)}} \leq \hat{\mu}_j + \sqrt{\frac{8\log t}{n_j(t)}}; a_t = j\right]$$

$$\leq \frac{2}{t^4} + (1 - \frac{2}{t^4})\mathbb{P}\left[\mu_i \leq \mu_j + 2\sqrt{\frac{8\log t}{n_j(t)}}; a_t = j\right], \text{[Due to Eq. (8)]}$$

$$\leq \frac{2}{t^4} + (1 - \frac{2}{t^4})\mathbb{P}\left[n_j(t) \leq \frac{32\log t}{\Delta_{ij}^2}; a_t = j\right] + \mathbb{P}\left[\mu_i \leq \mu_j + 2\sqrt{\frac{8\log t}{n_j(t)}}; n_j(t) > \frac{32\log t}{\Delta_{ij}^2}\right]$$

$$= \frac{2}{t^4} + (1 - \frac{2}{t^4})\mathbb{P}\left[n_j(t) \leq \frac{32\log t}{\Delta_{ij}^2}; a_t = j\right], \text{[Due to } \mu_i \geq \mu_j + \Delta_{ij}].$$

The above approach is standard in the analysis of the UCB based algorithms.

- If any arm $i$ is played $n_i(t)$ times upto time $t$ then it is available for $(t - n_i(t)D_i)$ time slots. Among these time slots where arm $i$ is available, UCBG can play

1) arms $1 \leq j \leq (i-1)$, at most $\sum_{j=1}^{(i-1)}\left(\frac{t}{D_j} + 1\right)$ times in total, w.p. 1, due to the blocking constraints;

and

2) the arms $(i+1) \leq j \leq K$, can be played at most $\sum_{j=(i+1)}^{K}\frac{32\log t}{\Delta_{ij}^2}$ many times in total, w.p. at least $(1 - K/t^3)$, due to the UCB property and union bound over all arms and time slots upto $t$.

Therefore, for all $i \leq K$ we have, w.p. at least $(1 - K/t^3)$,

$$n_i(t) \geq \frac{t}{D_i}\left(1 - \sum_{j=1}^{(i-1)}\frac{1}{D_j}\right) - \frac{1}{D_i}\left(\sum_{j=(i+1)}^{K}\frac{32\log t}{\Delta_{ij}^2} + (i - 1)\right). \tag{12}$$

More importantly, w.h.p. for all $i \leq K^*$ we see $n_i(t)$ grows linearly with time $t$. The above property quantifies the forced exploration in the system.

**Upper Bound on Optimization (9).**

From equation (12) we can infer that for each pair of arms $(i, j)$, $j \leq K^*$, and $i < j$, there exists an appropriate constant $\tau_{ij}$ such that after $\tau_{ij}$ timeslots we have $n_j(t) > \frac{32\log t}{\Delta_{ij}^2}$ w.p. at least $(1 - K/t^3)$. More specifically, we have for all $(i, j)$, $j \leq K^*$, and $i < j$, $\tau_{ij} \leq \tau_{(ij,0)}(1 + \log \tau_{(ij,0)}) + \tau_{(ij,1)}$,

$$\tau_{(ij,0)} = 32\left(1 - \sum_{l=1}^{(j-1)}\frac{1}{D_l}\right)^{-1}\left(\frac{D_j}{\Delta_{ij}^2} + \sum_{l=(j+1)}^{K}\frac{1}{\Delta_{jl}^2}\right), \quad \tau_{(ij,1)} = \left(1 - \sum_{l=1}^{(j-1)}\frac{1}{D_l}\right)^{-1}(j - 1).$$

The above follows using the relation $x > a\log x + b$, for all $x \geq (a\log a + a + b)$ given $a > e^{(e-1)}$.

Therefore, we can upper bound the regret as

$$\sum_{i=1}^{K_g}\sum_{t\in\mathcal{T}_i}\sum_{j=(i+1)}^{K} P_{ij}(t)\Delta_{ij} < \sum_{i=1}^{K_g}\sum_{t\in\mathcal{T}_i}\sum_{j=(i+1)}^{K}\Delta_{ij}\left(\tfrac{2}{t^4}+\mathbb{P}(n_j(t)\le\tfrac{32\log t}{\Delta_{ij}^2};a_t=j)\right)$$

$$\le \sum_{i=1}^{K_g}\sum_{t\in\mathcal{T}_i}\sum_{j=(i+1)}^{K}\tfrac{2\Delta_{ij}}{t^4}+\sum_{i=1}^{K_g}\sum_{j=(i+1)}^{K^*}\Delta_{ij}\sum_{t\in\mathcal{T}_i}\mathbb{P}(n_j(t)\le\tfrac{32\log t}{\Delta_{ij}^2};a_t=j)$$

$$+\sum_{i=1}^{K_g}\sum_{j=(K^*+1)}^{K}\Delta_{ij}\sum_{t\in\mathcal{T}_i}\mathbb{P}(n_j(t)\le\tfrac{32\log t}{\Delta_{ij}^2};a_t=j)$$

$$\overset{(i)}{\le}\sum_{i=1}^{K_g}\sum_{t\in\mathcal{T}_i}\sum_{j=(i+1)}^{K}\tfrac{2\Delta_{ij}}{t^4}+\sum_{i=1}^{K_g}\sum_{j=(K^*+1)}^{K}\tfrac{32\log T}{\Delta_{ij}}+\sum_{i=1}^{K_g}\sum_{j=(i+1)}^{K^*}\Delta_{ij}\sum_{t\in\mathcal{T}_i}\mathbb{P}(n_j(t)\le\tfrac{32\log t}{\Delta_{ij}^2};a_t=j)$$

$$\overset{(ii)}{\le}\sum_{i=1}^{K_g}\sum_{t\in\mathcal{T}_i}\sum_{j=(i+1)}^{K}\tfrac{2\Delta_{ij}}{t^4}+\sum_{i=1}^{K_g}\sum_{j=(K^*+1)}^{K}\tfrac{32\log T}{\Delta_{ij}}+\sum_{i=1}^{K_g}\sum_{j=(i+1)}^{K^*}\Delta_{ij}\sum_{t\in\mathcal{T}_i}\left(\tfrac{K}{t^3}+\mathbb{1}(t\le\tau_{ij})\right)$$

$$\overset{(iii)}{\le}\sum_{i=1}^{K_g}\left(2H(4)\tfrac{\mu_i-\mu_K}{D_i^4}+H(3)K\tfrac{\mu_i-\mu_{K^*}}{D_i^3}+\sum_{j=(i+1)}^{K^*}\tfrac{\Delta_{ij}\tau_{ij}}{D_i}\right)+\sum_{i=1}^{K_g}\sum_{j=(K^*+1)}^{K}\tfrac{32\log T}{\Delta_{ij}}.$$

Here, the inequality $(i)$ is true by noting $\sum_{t\le T}\mathbb{1}(n_j(t)\le\tfrac{32\log t}{\Delta_{ij}^2};a_t=j)\le\tfrac{32\log T}{\Delta_{ij}^2}$. the inequality $(ii)$, similarly follows with the additional use of the lower bound on $n_j(t)$ in Eq. (12). The inequality $(iii)$ follows by expressing $t\in\mathcal{T}_i$ as $lD_i$ for integers $l\ge1$, and then performing the summations. $\square$

**Remark.** Focusing on the $\sum_{j=(i+1)}^{K^*}\tfrac{\Delta_{ij}\tau_{ij}}{D_i}$, we observe that $\tau_{ij}=\tilde{\Theta}\left((1-\sum_{l=1}^{j-1}\tfrac{1}{D_l})^{-1}\right)$. Therefore, the constant term can become very large if $(1-\sum_{l=1}^{K^*-1}\tfrac{1}{D_l})$ is very small (even $\mathcal{O}(2^{-K^*})$ is possible).[4]

To avoid such large constants, alternatively we can substitute $K^*$ in the regret bound with the set $K_\epsilon^*:=\text{argmax}\left\{k:\sum_{i=1}^{(k-1)}\tfrac{1}{D_i}<1-\epsilon\right\}$. This will make the constant term in the regret bound $\tilde{\mathcal{O}}(\tfrac{K}{\Delta_{\min}\epsilon}+\tfrac{K^2\Delta_{\min}}{\epsilon})$, while worsening the $\log T$ dependence to $\sum_{i=1}^{K_g}\sum_{j=(K_\epsilon^*+1)}^{K}\tfrac{32\log T}{\Delta_{ij}}$.

*Proof of Theorem 4.1* By substituting $K^*$ in the regret bound with the set $K_\epsilon^* := \text{argmax}\left\{k:\sum_{i=1}^{k}\tfrac{1}{D_i}<1-\epsilon\right\}$ in the above proof we get back Theorem 4.1. $\square$

**Gap-Independent Bound.** Using standard approach [4] we obtain a gap-independent bound.

$$\sum_{i=1}^{K_g}\sum_{t\in\mathcal{T}_i}\sum_{j=(i+1)}^{K} P_{ij}(t)\Delta_{ij} < \sum_{i=1}^{K_g}\sum_{t\in\mathcal{T}_i}\sum_{j=(i+1)}^{K}\Delta_{ij}\left(\tfrac{2}{t^4}+\mathbb{P}(n_j(t)\le\tfrac{32\log t}{\Delta_{ij}^2};a_t=j)\right)$$

$$\le T\Delta+\sum_{i=1}^{K_g}\sum_{t\in\mathcal{T}_i}\sum_{j:\Delta_{ij}>\Delta}^{K}\Delta_{ij}\left(\tfrac{2}{t^4}+\mathbb{P}(n_j(t)\le\tfrac{32\log t}{\Delta_{ij}^2};a_t=j)\right)$$

$$\le T\Delta+\sum_{i=1}^{K_g}\tfrac{2H(4)}{D_i^4}+\sum_{i=1}^{K_g}\sum_{j=i}^{K}\tfrac{32\log T}{\Delta}$$

Substituting, $\Delta=\tfrac{\log T}{K_g KT}$ we obtain the gap independent bound of $\mathcal{O}(\sqrt{K_g KT\log T})$.

# D Proof of Regret Lower Bound

We make two key observations regarding the behavior of the two algorithms in the special case when all the delays are equal, say $D < K$. Firstly, in this setting, the optimal algorithm plays the $D$ best arms in a round robin manner following the cycle $\{1, 2, \ldots, D\}$. Furthermore, it is easy to see that the Oracle Greedy coincides with the optimal algorithm.

Secondly, for equal delay system the feedback received by any online algorithm is identical to the so called semi-bandit feedback [17, 11]. Specifically, consider the alternative system where the time horizon is partitioned into contiguous blocks of length $D$ each block acting as a new time slot. In each new time slot/block, $D$ distinct arms are played and the instantiation of the individual rewards of these $D$ arms become visible. This is a well studied problem known as combinatorial semi-bandit [24]. The rest of the proof first makes the connection to combinatorial semi-bandit rigorous and then follows an mapping to Bernoulli bandits (the latter is similar to the lower bound in [24].)

**Lemma D.1.** *For any Blocking Bandit instance where $D_i = D \leq K$ for all arms $i \in [K]$, time horizon $T$, and any online algorithm $\mathcal{A}_O$, there exists an online algorithm $\mathcal{A}_B$ which chooses arms for blocks of $D$ time slots and obtain the same distribution of the cumulative reward as $\mathcal{A}_O$.*

*Proof of Lemma D.1.* We prove the above by induction for each sample path separately. We fix an arbitrary online algorithm $\mathcal{A}_O$. We construct an online algorithm which is forced to choose arms for blocks of $D$ time slots each, namely $\mathcal{A}_B$ to simulate $\mathcal{A}_O$ in the semi-bandit feedback. Specifically, let $I_t$ be the arm played at time $t$ by $\mathcal{A}_O$. The belief on the reward of arm $i$ at the beginning of time $t \geq 1$, namely $\mathbb{P}_i(t)$, is a function of the instantiations of the arm seen so far, $\{X_i(t') : t' \leq (t-1), I_{t'} = i\}$. As our objective is to prove equality of cumulative reward distribution, due to the i.i.d. nature of the rewards we can restrict ourselves to $\mathcal{A}_O$ given by the sequence $I_t : \{\mathbb{P}_i(t) : i \in A_t\} \to A_t$ ($A_t$ are the available arms in time slot $t$).

We observe that for all $i \in A_t$, we gain no information in time $(t - D)$ to $t$, as it can not be played due to blocking constraint, i.e. $\{X_i(t') : t' \leq (t-1), I_{t'} = i\} = \{X_i(t') : t' \leq (t-D), I_{t'} = i\}$. This implies for all $i \in A_t$, $\mathbb{P}_i(t) = \mathbb{P}_i(t'), \forall (t-D) \leq t' \leq t$ (same distribution). Therefore, if we divide the time slots into blocks of length $D$, we have

$$\forall j \geq 0, \forall (jD+1) \leq t \leq (j+1)D; \{\mathbb{P}_i(t) : i \in A_t\} \subseteq \{\mathbb{P}_i(jD+1) : i \in [K]\}.$$

The above argument shows that it is sufficient to consider $\mathcal{A}_O$ which is given by the sequence $I_t : \{\mathbb{P}_i(jD+1) : i \in [K]\} \to A_t, \forall j \geq 0, \forall (jD+1) \leq t \leq (j+1)D$. However, this is indeed an online algorithm $\mathcal{A}_B$ which chooses arms $\{I_t : (jD+1) \leq t \leq (j+1)D\}$ in the beginning of the $j$-th block (i.e. on $jD$-th time slot). This proves our claim. $\square$

*Proof of Theorem 4.3.* Let us now consider the instance with $K$ arms each with delay $K^* < K$. Let the reward of the arms $i = 1$ to $K^*$ be distributed as Bernoulli distribution with mean $0.5$. For the arms $i = (K^* + 1)$ to $K$ the rewards are distributed as Bernoulli distribution with mean $(0.5 - \Delta)$. Due to Lemma D.1, we can reduce this problem to the bandits with multiple play problem [1], where in each block we can play $K^*$ distinct arms. The regret is lower bounded for this problem by $\sum_{i=(K^*+1)}^{K} \frac{\Delta}{D_{KL}(0.5||0.5-\Delta)}$, where $D_{KL}(p||q)$ is the Kullback-Leibler divergence between Bernoulli distributions. We can bound $\frac{\Delta}{D_{KL}(0.5||0.5-\Delta)} \leq \frac{1}{4\Delta}$, which completes the proof. $\square$

**Oracle Greedy for Exact Covering System.** Recall for the exact covering system under consideration, we know for any $i, j \in [K_g], i \neq j$ $seq_i \cap seq_j = \emptyset$ and $[T] = \cup_{i=1}^{K_g} seq_i$.

In this exact covering system the oracle greedy (with appropriate tie breaking) places arm $i$ (recall $\mu_i$ are non-decreasing w.r.t. $i$) at the positions $seq_i = \{i + nD_i : n \in \mathbb{N}\}$ for all $i \in [K_g]$. This can be proved inductively on $i$. Oracle greedy places arm $1$ in the positions $seq_1$, as arm $1$ is available. For $[T] \setminus \cup_{i=1}^{k} seq_k$ the best available arms are arm $k + 1$. Also $\cup_{i=1}^{k} seq_k \cap seq_{(k+1)} = \emptyset$. Therefore, oracle greedy will place arm $k + 1$ in positions $seq_{(k+1)}$.

It is easy to see, for any $T \geq 1$, the Oracle greedy achieves reward $\sum_{i=1}^{K^*} T \frac{\mu_i}{D_i} - \mathcal{O}(1)$. Indeed, this is the asymptotically optimal reward.

## Footnotes

[3]For any real numbers $a_i \in (0, 1)$, $\prod_{i=1}^{n}(1 + a_i) \geq 1 + \sum_{i=1}^{n} a_i$ and $\prod_{i=1}^{n}(1 + a_i) \geq 1 + \sum_{i=1}^{n} a_i$.

[4] $\tilde{\Theta}(\cdot)$ and $\tilde{\mathcal{O}}(\cdot)$ hides the logarithmic terms.