[Reviews · NeurIPS 2019]

Reviewer 1



The paper makes three key contributions. First, the authors provide results on hardness of the problem by reducing it to the PINWHEEL scheduling problem and provide negative results on solvability of the problem even with prior knowledge of the rewards and delays of all the arms. Second, motived by the obtained result on hardness of the problem, the authors show that a simple greedy algorithm that plays the available arm with the highest reward is asymptotically (1 -1/e) optimal. Finally, the authors show that by exploiting the free exploration of arms due to the unavailability the UCB algorithm achieves the log T cumulative regret with respect to the oracle greedy algorithm with known rewards. The presentation of the paper was mostly clear. The claimed contributions are discussed in the light of existing results and the paper does survey related work appropriately. Specifically, the authors put and compare the proposed setting in the context of other settings such as sleeping bandits, combinatorial semi-bandits, and bandits with budget. Regarding the quality of the writing, the paper is reasonably well written, the structure and language are good. The paper is technically sound, and the proofs seem to be correct as far as I checked.

Reviewer 2



This paper introduces an interesting and original framework for multi-armed bandits with potential applications. I liked the impossibility result and the global approach of the paper. Yet, in my opinion, the paper still needs some more polishing. It sometimes lacks clarity and contains some typos or proofs that I found hard to follow. Furthermore, the tightness of the results should maybe better discussed. I have two main comments: - The lower bound and the upper-bound do not really match. In particular, if we apply the reward instance used in the lower bound to the upper bound, the latter becomes infinite (because the Delta_{ij} are zero), am I right? I found the dependence on 1/Delta(K,K^*) quite weak since this quantity may be really small. I would be interested to see a lower-bound with that dependence or at least experiments to see if this is really observed in practice. Furthermore, there is also a gap of a factor K with between the upper-bound and the lower-bound. - What about the dependence on the delays? The proposed algorithm does not use knowledge of the delays. Would it be possible to improve the results if they are known or not? The impossibility result states that we cannot reach the optimal strategy but can we get closer to it? Other Remarks: - Distribution-free upper-bound. The quantity Delta(K,K^*) may indeed be very small in many situations, is it possible to state also a distribution-free bound? - Is the factor (1-1/e) tight? The best factor obtained by the lower bound is 3/4, isn't it? - About Prop 3.4: - The choice epsilon=0 seems to be the best leading to a ratio 3/4. Why do you need epsilon at all? - As in all the proposition, the statement should be more accurate. In particular, it is assumed epsilon<1 which is not stated in the prop. Furthermore, "for any epsilon" should be before "There exists" since for each epsilon the instance is different. - About Prop 3.5: I could not follow at all the proof. The greedy algorithm plays the (K+1)th arm if (1+eps)/(K+1)>1 but then its cumulative reward is better than the one stated for the optimal algorithm... What did I miss? - About Thm 4.1: the upper-bound is a bit hard to read. It would be useful to simplify it or to explain the different terms. - In the experiments, it is stated that sometimes, the greedy algorithm gets blocked in a wrong cycle leading to a negative cumulative regret. I would expect that the opposite is also possible, with UCB blocked in a suboptimal cycle. Isn't it possible with small probability? A high-probability regret bound would be useful. Furthermore, would some random perturbation of the prediction be possible to try to fall into a better cycle? - Proof of the examples l.254 would be nice - I do not understand how K^*=20 is possible in the experiments since the delays are all smaller than 20. Are all delays equal? Typos: - l.72: imu2>...>muK should be stated before the definition of Delta - l.73: >=1 missing in the definition of K^* - In some results, there are K arms and some other K+1 arms. Please make it uniform to ease the reading. - l.205: note that H is the Riemann zeta function - l.208 and later: Delta_{ij} -> Delta(i,j) otherwise define it - l.236: ",," - l.237: exploraiton

Reviewer 3



The paper studies a version of multi-armed bandit in which when arm i is pulled, it cannot be pulled again for at least d_i rounds. While the problem itself is interesting, I have some doubts about the results. My most important concern is about the lower bound for the offline version of the problem. I am suspicious about the reduction from this problem, named MAXREWARD, to another problem called PINWHEEL SCHEDULING (the reduction presented in the appendix). Here is why: an instance of PINWHEEL SCHEDULING turns to an instance of MAXREWARD with size \Pi_i d_i = d_1 * ... * d_k (because T has to be this large in order to distinguish YES cases from NO cases). On the other hand, it is not hard to see PINWHEEL SCHEDULING does admit a dynamic programming algorithm whose running time is polynomial in \Pi_i d_i = d_1 * ... * d_k. Therefore I am not sure what conclusion this reduction is supposed to make. ---- UPDATE AFTER REBUTTAL: Here is more details on my comment and my confusion regarding the lower bound: For the lower bound, a decision version of the problem in the following form is studied: "Given an instance of the The PINWHEEL SCHEDULING and T, is the optimal reward T?" and it is assumed that T is given in binary format so a "poly running time" is expected to be poly in log(T). ---- Also there are minor issues in the paper which makes it a bit hard to read. One of them is in the definition of \tau_{i, t}. Is it defined correctly? I guess the authors actually meant to replace min_{t' >= 1} with max_{t'<= t}

Reviewer 4



The paper studies a new multi-armed bandit setup, where after each pull of an arm, it is blocked (i.e. zero reward) for a certain number of rounds (this number is known). The setup captures some real-life applications nicely as motivated in the beginning of the paper. The paper also makes it clear in Sec 1.2 how this new setup is connected to existing ones and why those results do not directly solve the problem. The paper includes a set of theoretical results as listed above, starting from the natural question of if one can solve the offline version of the problem with knowledge of expected rewards of all arms, and then coming back to the online setting and discussing how to ensure low regret against a reasonable offline strategy. The techniques of proving the regret bound (highlighted on Page 2) indeed seem novel. There are still several loose ends though. For example, the regret upper and lower bounds do not match exactly. Also, it is not entirely clear to me that a computational hardness result for finding the exact solution of the offline problem excludes the possibility of efficient online algorithm with sublinear (exact) regret. Overall, I believe the paper studies an interesting new setting and provides several solid results. I recommend accept.

[Author Response · NeurIPS 2019]

We thank the reviewers for their insightful comments, which have helped us improve the paper. We now discuss the
issues raised by the reviewers and our proposed changes in the final version to resolve these. We omit discussion for
each suggested change which we will directly implement in the final version.

**Simplified Regret Upper Bound:** We define $D_{min} := \min_i D_i$, $D_{max} := \max_i D_i$, and $\Delta_{i,i+1} := (\mu_i - \mu_{i+1})$.
The regret admits the bound $R_T^{(1-1/e)} \le O(\frac{1}{\epsilon} \log(\frac{1}{\epsilon})) + \frac{32 K_g (K - K_\epsilon^*)}{\min_{i \in [K_\epsilon^*, \ldots, K_g]} \Delta_{i,i+1}} \log(T)$ for any $\epsilon > 0$. Notably $\Delta_{i,i+1}$
for $i = 1$ to $(K_\epsilon^* - 1)$ do not appear in the bound. Furthermore, $(1 - \epsilon) D_{min} \le K_\epsilon^* \le \min(K, (1 - \epsilon) D_{max} + 1)$ for
any $\epsilon \ge 0$, and $D_{min} \le K_g \le \min(K, D_{max})$. Using these bounds we can simplify the regret upper bound.

**Reviewer #4:** • We will add the above simplified regret upper bound in the final version of the paper.

• The problem resolves to semi bandit only if *all the delays are equal*. When all the delays are less than $K$ (number of
arms) but distinct we do not have reduction to the semi-bandit case. Nonetheless, we simulate the results for systems
where $\max_i D_i > K$ ($K = 20$ and $\max_i D_i = 40$) and observe similar behavior as reported in the current version of
the paper. We will report the results of the new experiments in the final version of the paper.

**Reviewer #5** • The upper bound for the $K^*$-set instance (the instance used in the lower bound proof), is $O(1) +$
$\frac{K^*(K - K^*) \log(T)}{\Delta}$. This follows by choosing any $0 < \epsilon < 1/K^*$, using $K_g = K^*$, and observing that for all
$i \in \{1, \ldots, K^*\}$ and $j \in \{K^* + 1, \ldots, K\}$, $\Delta_{ij} = \Delta$.

• The lower bound is $\Omega((K - K^*)/\Delta(K, K^*))$ as proved in the paper. Therefore, the dependence on $\Delta(K, K^*)$ is
statistically optimal. However, the lower bound is smaller than the upper bound by a factor of $K_g \le \min(K, D_{max})$.
Closing this gap is left as future work.

• We would like to emphasize here that a key message of our paper is that in the offline setting the simple greedy
algorithm is (1-1/e) optimal. The goal of the next result is to show that the UCB greedy algorithm has low regret *w.r.t*
*the greedy algorithm*, and the delay information is not required for this result. This does not rule out the possibility of
improved algorithms that use the delay information, both in the offline and online settings.

• A gap-independent regret upper bound of $2H(4)/(D_{min})^4 + \sqrt{32 K_g K T \log(T)}$ holds. We will add the result and
its proof in the supplementary material for the sake of completeness.

• The use of $\epsilon$ in Prop. 3.4 is purely technical to avoid tie breaking in favor of arm 3. Indeed, if arm 3 has reward 1,
same as arm 1 and arm 2. The greedy algorithm will play 13231323… which gives the optimal reward. However, for
any $\epsilon > 0$ greedy plays suboptimally as 12341234… Therefore, using $\epsilon > 0$ is required for the proof. We will make
the statement of Prop 3.4 rigorous by swapping the order of "there exists" and mentioning $\epsilon < 1$.

• Prop 3.5 states suboptimality of an algorithm called the greedy-per-round which plays the available arm with the
highest "reward/delay". Here, we have $(1 + \epsilon)/(K + 1) > 1/(K + 1)$, not $(1 + \epsilon)/(K + 1) > 1$. We will explain the
greedy-per-round algorithm clearly and explicitly mention the inequality $(1 + \epsilon)/(K + 1) > 1/(K + 1)$.

• The UCB-greedy algorithm may enter suboptimal cycles but as high reward arms become available, UCB strategy
ensures they are played with high probability. We will consider the high probability regret upper bound as a future work.

• For Fig. 1.b, where $K^* = 20$, all delays are equal. Here we highlight the constant regret behavior when $K^* = K$.

**Reviewer #6** • The hardness result stated in the paper is with respect to the unary representation of the number of
arms, i.e. time complexity is measured w.r.t. the number of arms, K. Specifically, the number of bits required to
express the total number of timeslots T can be expressed in $\log(\Pi_i D_i)$ bits which is polynomial in the number of
arms. Now recall that the PINWHEEL scheduling under consideration is dense. Therefore, by construction, for the
MAXREWARD instance the delays will satisfy $\sum_{i=1}^K 1/D_i = 1$. Now the decision problem for the MAXREWARD
becomes "$OPT = T \sum_{i=1}^K a_i/D_i$?" Thus the decision problem is expressible in polynomial number of bits. Therefore,
the reduction is valid. We will add the above explanation in the proof.

• We thank the reviewer for pointing out this key typographical error. We will replace $\min_{t' >= 1}$ with $\max_{t' <= t}$.

• Our focus in terms of the offline problem is studying the complexity of the problem both in negative and positive
direction. Specifically, we prove the hardness of the optimization problem (in number of arms), and in the positive side
we show that the greedy heuristics under full information is a (1-1/e) approximation. Indeed, as pointed out by the
reviewer, we leave open the design of an algorithm with approximation larger than (1-1/e) [or proving approximating
beyond (1-1/e) is hard]. We will highlight this message clearly and specify the future works.

• We have checked the existing literature very carefully and we believe that the paper introduces and studies a new
problem. Furthermore, the other reviews also acknowledge the novelty of the problem. Therefore, we request the
reviewer to re-evaluate the work, as mentioned at the end of the review.

[Meta-Review · NeurIPS 2019]

The paper proposes a new bandit setup that is theoretically interesting and practically relevant. The reviewers generally agree that the results are solid, although there is still room for improvement in terms of both theory and the presentation. Please do take all the reviewers' comments into account when preparing the final version. Note: one of the main complains from Reviewer #6 is that the problem seems to have been studied before, but the reviewer provided no references. To the best of the knowledge of the other reviewers and the AC, the problem is new. Reviewer #6 promised to increase the score if this is indeed the case, but was not responsive at all during the discussion. Therefore, this review was not taken into account after all.